# Adjusting Machine Learning Decisions for Equal Opportunity and Counterfactual Fairness

**Yixin Wang**                                                                    *yixinw@umich.edu*
*University of Michigan*

**Dhanya Sridhar**                                                    *dhanya.sridhar@mila.quebec*
*Mila-Quebec AI Institute and Université de Montréal*

**David M. Blei**                                                          *david.blei@columbia.edu*
*Columbia University*

*Reviewed on OpenReview:* `https://openreview.net/forum?id=P6NcRPb13w`

## Abstract

Machine learning (ml) methods have the potential to automate high-stakes decisions, such as bail admissions or credit lending, by analyzing and learning from historical data. But these algorithmic decisions may be unfair: in learning from historical data, they may replicate discriminatory practices from the past. In this paper, we propose two algorithms that adjust fitted ML predictors to produce decisions that are fair. Our methods provide post-hoc adjustments to the predictors, without requiring that they be retrained. We consider a causal model of the ML decisions, define fairness through counterfactual decisions within the model, and then form algorithmic decisions that capture the historical data as well as possible, but are provably fair. In particular, we consider two definitions of fairness. The first is "equal counterfactual opportunity," where the counterfactual distribution of the decision is the same regardless of the protected attribute; the second is counterfactual fairness. We evaluate the algorithms, and the trade-off between accuracy and fairness, on datasets about admissions, income, credit, and recidivism.

## 1 Introduction

There is growing interest in using machine learning (ml) methods to automate important decisions about people by analyzing and learning from historical data. For example, in the criminal justice system, ml algorithms are routinely used to assess a defendant's risk of recidivism to inform pretrial release and parole decisions (Brennan et al., 2009; Larson et al., 2016; Lin et al., 2020). ml algorithms were also used by a university to help determine which college applicants should be reviewed (Waters & Miikkulainen, 2014).

As a running example, consider an admissions committee that decides which applicants to accept to a university. Given historical data about applicants' traits and the admission decisions, an ml algorithm can learn to predict who will be admitted and who will not, and the resulting ml decision-maker will accurately simulate the committee's decisions. But while these algorithmic decisions would save time and effort, they will also inherit some of the undesirable properties of the admissions committee (Corbett-Davies et al., 2017; Mitchell et al., 2021). If the committee was unfairly biased then its ml replacement will be as well.

In this paper, we develop two methods that adjust existing ML decision-makers to produce algorithmic decisions that are both accurate and fair. We prove that these algorithmic decisions maintain as much fidelity as possible to the decisions of the historical committee, but are adjusted to correct for the biases inherent in past decisions. We demonstrate these algorithmic decisions on datasets about income, credit, and recidivism.

A key contribution of the proposed methods is that they do not require refitting the ML method to data. One can train an ML decision-maker as usual, but then use the proposed methods to adjust the resulting

algorithmic decisions to be fair. Consequently, these methods can be employed by practitioners who have a pre-trained ml model but without the original data, or when the ml model is too expensive to refit.

**Main idea.** To quantify the fairness of a decision maker, we use two causal criteria. Causal criteria for fairness were first introduced in Kusner et al. (2017), and have been operationalized in many algorithms (Zhang & Bareinboim, 2018a;b; Zhang et al., 2017; Wu et al., 2019a;b; Coston et al., 2020; Kilbertus et al., 2017; 2019; Chiappa, 2019). Causal fairness criteria require encoding our assumptions about bias using causal models. While these assumptions require careful scrutiny (Kohler-Hausmann, 2018), causal models enable us to derive fairness measures to capture different mechanisms of bias. In contrast to causal criteria, statistical measures of fairness cannot distinguish between different mechanisms of unfairness (Hardt et al., 2016).

First, we introduce the equal counterfactual opportunity (eco) criterion. With an eco decision-maker, individuals that have the same attributes will receive the same probability of admission, regardless of their demographic group. In particular, for an algorithmic decision that satisfies eco, there is no causal relationship between the protected attributes (such as race or gender) and the decision. The eco criterion is individual-level; it requires that each decision is fair.

Second, we consider the counterfactual fairness (cf) criterion proposed in Kusner et al. (2019). Under cf, the probability of admission is not affected by (hypothetical) interventions on protected attributes, including when we account for how those interventions might subsequently change the other attributes. For example, minority applicants may have been historically denied university admission due to lower test scores, but these low scores might be consequences of systemic disadvantages that the minority applicants face. A cf decision-maker adjusts an individuals' attributes (such as test score) to account for such disadvantages. Similar to the eco criterion, the cf is also individual-level.

With these two criteria in place, the methods we develop draw on the ideas and methods of causal inference (Pearl, 2009; Peters et al., 2017). First we require that the historical decision-making process is captured by a causal model. This model reflects domain knowledge and assumptions about the process, including the probabilistic relationship between an individual's attributes and the decision, and relationships between the attributes themselves. The causal model might be elicited from experts, informed from data, or a combination.

Continuing the example, a causal model of admissions represents the applicants' traits and the committee's decisions. In this paper we use the model illustrated in Fig. 2. There are protected attributes, such as gender, and non-protected attributes, such as test score. The model posits that the (historical) decision was affected by both, e.g., the committee considered the test score but also unfairly considered gender. The model also posits that the non-protected attributes may be affected by the protected ones, e.g., gender may affect the test score through systematic gender bias in schooling. The methods we develop do not require this particular causal model, but it will be a running example.

Given the causal model of historical decisions, we then frame *algorithmic* decisions as new causal variables in the same model, ones whose values are functions of the individuals' attributes. This framing is important because it allows us to consider counterfactuals about the algorithms, e.g., "what would the algorithm decide for this applicant if she was male and had achieved the same test score?" Probabilistic properties of such counterfactuals help formally define the mathematical criteria for eco and cf.

Finally, we derive methods to produce algorithmic decisions that meet these criteria. The first step is to use the historical data to fit an ml method to predict the decision, such as with a simple logistic regression or a neural network. (Note we can then use the causal model and causal criteria to assess the fairness of the ML decision-maker.) We then show how to adjust the trained ml decision-maker to satisfy eco, and we show how to adjust the eco decision-maker to exercise cf. We will see that both adjustment procedures involve estimating causal quantities based on both the assumed causal model and the original ml predictions.

We emphasize again that this method uses a fitted ml decision-maker but does not require retraining it or re-analyzing the historical data, which could require significant computational cost. We prove that, under the assumed causal model, the fair decision-makers we derive are theoretically optimal—they maintain as much fidelity to the historical data as possible, while still being eco-fair or cf-fair.

We study these approaches on simulated admissions data and on three public datasets, about income, credit, and recidivism. We find that classical ml decisions are the most accurate, but they may be unfair. The fair decision-makers deviate from the ml decisions, but provide decisions that meet the fairness criteria. Compared to other approaches that satisfy eco and cf (Chen et al., 2019; Kusner et al., 2017), the method developed here provides decisions with higher accuracy while remaining fair.

**Contributions.** 1) We develop algorithms that modify existing ml predictors to be fair, without requiring retraining. 2) We prove that these decisions optimally recover ml predictors while satisfying the criteria for eco and CF. 3) We conduct empirical studies on real and simulated settings to show that our algorithms produce better decisions than existing approaches that satisfy the same fairness criteria.

**Related work.** This paper contributes to research about causal models for defining and implementing fair ML. Kusner et al. (2017) introduce counterfactual fairness (the cf criterion above) and the FairLearning algorithm to satisfy it. This paper provides eco-fair decisions and cf-fair decisions that minimally adjust the fitted ml decisions. Like FairLearning Kusner et al. (2017), the cf-fair decisions also satisfy counterfactual fairness. But in contrast to FairLearning, which omits descendants of the protected attributes, the cf decision-maker in this paper uses all available attributes and achieves higher fidelity to historical decisions.

Kilbertus et al. (2017) considers more fine-grained causal notions of fairness and relates paths and variables in causal models to violations of eco and cf. Other work studies path-specific fairness, where some paths in the causal model are deemed unfair. These works typically propose new training objectives that satisfy fairness (Chiappa, 2019; Zhang et al., 2017; Nabi & Shpitser, 2018; Zhang & Bareinboim, 2018b; Wu et al., 2019b; Mhasawade & Chunara, 2021; Dutta et al., 2021; Madras et al., 2019). In contrast to the methods developed for path-specific fairness, this paper targets individual-level fairness, where each decision is guaranteed to be fair. Moreover, the algorithmic decisions we propose in this paper do not require re-training the ml decision-makers.

A number of recent papers further extend the idea of counterfactual fairness. Kusner et al. (2019); Mishler & Kennedy (2021) develop decision-makers that target multiple decisions or multiple fairness criteria. Coston et al. (2020) extends the use of counterfactuals for fair risk assessment scores. Galhotra et al. (2021); Parafita & Vitria (2021); Dai et al. (2021); Foster et al. (2021); Artelt et al. (2021); Karimi et al. (2021); Veitch et al. (2021) develop algorithms for counterfactual invariance and counterfactual fairness in model explanation and representation learning. Mishler et al. (2021) extends counterfactual fairness to counterfactual equalized odds and propose post-processing optimization programs for adjustment. Unlike these works, this paper focuses on developing post-hoc fairness adjustments with closed forms, without requiring retraining.

Alongside these causal approaches to algorithmic fairness, there is a vast literature on statistical criteria for fairness. We refer the readers to Mitchell et al. (2021); Corbett-Davies & Goel (2018); Pessach & Shmueli (2022); Berk et al. (2021) for surveys of these related ideas.

The literature on fair ML contrasts fairness criteria that apply to each individual and those that apply to groups. The definitions of eco and cf in this paper are individual-level criteria. Similarly, Dwork et al. (2012) define a notion of eco without causality called individual fairness. Given a distance metric between individuals, it requires similar individuals receive similar decisions. The eco-fair decisions (of Eq. 1) recover this requirement without relying on an explicit distance metric, which can be difficult to construct. Speicher et al. (2018); Chouldechova & Roth (2019) study the tradeoffs between group- and invididual-level criteria.

Finally, the method proposed here crucially relies on the theory and algorithms behind causal inference (Pearl, 2009; Peters et al., 2017). In particular, it requires that the causal model is accurate, that the necessary counterfactuals are identifiable, and that they can be estimated. In practice, these assumptions are strong and have the potential to be misused (Kasirzadeh & Smart, 2021; Kohler-Hausmann, 2018). To alleviate some of this burden, subsequent research can extend this method to employ recent ideas about causal fairness under uncertain causal models (Kilbertus et al., 2018; Russell et al., 2017; Kilbertus et al., 2019; Wu et al., 2019a;b; De Lara et al., 2021).

## 2 Assessing fairness with counterfactuals

Consider automating the admissions process at a university. By using a dataset of past admissions, the goal is to algorithmically compute the admissions decision for new applicants. The dataset contains $n$ applicants, each with attributes and the committee's decision about whether to admit them. Some attributes are deemed protected, such as gender, religion, or ethnicity; others are not protected, such as a grade point average or a score on a standardized test.

To illustrate the ideas, we will consider a simple setting. Suppose there are only two attributes for each applicant, their gender and their score on a test. (For ease of exposition, we consider two attributes and a binary gender variable, but the methods in this paper easily handle multiple attributes and non-binary gender.) The gender is a protected attribute; the test score is not. Fig. 1 shows an example of such data. In the figure, the first 5,000 applicants are (simulated) historical data—applicants' attributes and the decisions made by a committee. The three applicants A, B, and C are new applicants for whom we will make decisions.

**A causal model of the decisions.** The methods we develop require a causal model (Pearl, 2009) of the historical admissions process. This model should capture the decision, the inputs to the decision, and the causal relationships among these variables. Consider the model in Fig. 2. It assumes that an applicant's gender $A$ and test score $S$ can affect the admissions decision $Y$; the admissions committee might unfairly prefer to admit males. It also assumes the applicant's gender $A$ can affect the test score $S$; female applicants might receive fewer opportunities for test preparation, a disparity that results in lower scores. (There are also other variables in Fig. 2, which we will discuss below.) Note that positing a causal model is a strong assumption; it is important for the user of the methods to consult domain knowledge and provide a plausible model of the historical decisions. Many causal models are special cases of the causal model in Fig. 2, if we have multiple (or multi-dimensional) protected and unprotected attributes. We discuss these cases in § 3.

A causal model implies a distribution of *counterfactuals*, how each variable would change under hypothetical intervention on other variables (Pearl, 2009). (Note this "intervention" is only a mathematical construction, a hypothetical change that helps articulate counterfactuals. It may be about an immutable characteristic (Greiner & Rubin, 2011).) Denote $Y_i(a, s)$ to be the counterfactual decision of the $i$th applicant when we set their test score to be $S_i = s$ and gender to be $A_i = a$. For example, consider the counterfactual decision of a female applicant had she been a male and had the same test score. A key idea of this paper is that we can use the assumed model to ask counterfactual questions about any decisions, either those made by a committee or those made by an algorithm. We first show how we can interpret fairness through properties of such counterfactual decisions. We then show how to adjust an algorithmic decision maker, such as one produced by classical ml, to produce decisions that satisfy fairness properties.

**The equal counterfactual opportunity criterion.** A decision maker provides equal counterfactual opportunity (eco) when equally qualified people receive the same decisions regardless of their protected attributes. In the causal model, we say that a decision

| ID | Sex | Test | Admit | $\hat{Y}^{\mathrm{ml}}$ | $\hat{Y}^{\mathrm{eco}}$ | $\hat{Y}^{\mathrm{cf}}$ |
|---|---|---|---|---|---|---|
| 1 | f | 54 | yes | | | |
| 2 | m | 66 | no | | | |
| $\vdots$ | $\vdots$ | $\vdots$ | $\vdots$ | | | |
| 5000 | f | 44 | no | | | |
| A | f | 85 | ? | 0.67 | 0.77 | 0.78 |
| B | m | 85 | ? | 0.84 | 0.77 | 0.76 |
| C | f | 65 | ? | 0.57 | 0.69 | 0.70 |

**Figure 1:** Simulated data of past university applicants (1-5000) and their admissions decisions. The (fictional) admissions committee violates eco and does not enforce cf. For three new applicants (A, B, C), a machine learning classifier trained on this biased data yields unfair decisions, denoted by $\hat{Y}^{\mathrm{ml}}$. For example, a male and female applicant with the same test score receive different admission probabilities, violating eco. We algorithmically adjust this decision to produce $\hat{Y}^{\mathrm{eco}}$ decisions that are eco-fair. This decision provides the equally qualified two candidates A and B an equal probability of being admitted. From this $\hat{Y}^{\mathrm{eco}}$ decision, we derive $\hat{Y}^{\mathrm{cf}}$, an algorithmic decision that enforces cf. It increases the probability of admission for applicant C, by accounting for her systematic disadvantage.

maker satisfies eco if changing the protected attribute $A$ does not change its distribution of the decision $Y$. Consider an applicant with gender $a$ and test score $s$. An eco decision maker gives the same probability of admission even after changing the gender to $a'$ (but keeping the test score at $s$). In Fig. 1, candidates A and

B have the same test score but different genders; if given equal opportunities, they should receive the same probability of admission.

**Definition 1** (Equal counterfactual opportunity (eco)). *A decision $Y$ satisfies eco in the protected attribute $A$ if, for all possible values of $a$, $a'$ and $s$,*

$$Y(a,s) \,|\, \{A = a, S = s\} \stackrel{d}{=} Y(a',s) \,|\, \{A = a, S = s\}.$$

*(The notation $Q \stackrel{d}{=} R$ means $Q$ and $R$ are equal in distribution.)*

This criterion requires that, for any individual with protected attributes $A = a$ and unprotected attributes $S = s$, their counterfactual decisions $Y(a', s)$ must have the same distribution if they had the same unprotected attributes, no matter what the protected attribute value $A = a'$ may be.

An algorithmic decision that satisfies eco asserts that there is no causal relationship between the protected attributes $A$ and the decision $Y$. If a committee or algorithm produces eco-fair decisions then the causal model contains no arrow between $A$ and $Y$; two applicants from different demographic groups, but who are otherwise similarly qualified, will have the same probability of admission.

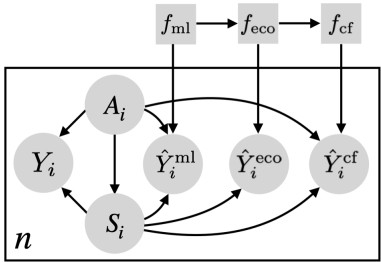

**Figure 2:** The causal model reflects domain knowledge and assumptions about an admissions decision process: protected attributes $(A)$, such as gender, can affect both the admissions decision $(Y)$ and the remaining attributes $(S)$, such as test scores. Algorithmic decisions $(\hat{Y}^{\mathrm{ml}}, \hat{Y}^{\mathrm{eco}}, \hat{Y}^{\mathrm{cf}})$ are variables in the causal model, produced by fitted algorithms $(f_{\mathrm{ml}}, f_{\mathrm{eco}}, f_{\mathrm{cf}})$ and we can inspect their biases. Classical ml reproduces the historical data it was trained on, inheriting the unfairness of a past admissions committee. eco decisions adjust ml, eliminating any problematic dependence on the protected attribute. cf decisions further adjust eco decisions to exercise affirmative action based on the protected attributes.

Other research has posited mathematical criteria to capture the legal notion of equal opportunity. Hardt et al. (2016) proposed the equality of opportunity criterion to evaluate the fairness of decisions predicted by a classifier. Their criterion requires the misclassification rates for the disadvantaged group to be the same as those for the other group. In contrast, the eco criterion is finer-grained; it requires that the decisions are fair under each possible configuration of the attributes that are input into the decision algorithm; all individuals with the same unprotected attribute must receive the same decision probability regardless of interventions on their protected attributes. Another fine-grained criterion appears in Dwork et al. (2012), which defines a notion of equal opportunity (without causality) called individual fairness. Given a distance metric between individuals, it requires that similar individuals receive similar decisions. Algorithmic decisions that satisfy the eco criterion also recover this requirement, but with a distance metric implied by the causal model.

Finally, eco criterion is also equivalent to conditional demographic parity (Dwork et al., 2012), conditional on all remaining attributes, if all remaining attributes (the unprotected attributes) are descendants of the protected attribute in the causal graph. For example, if the protected attribute is current monthly income, and the unprotected attributes include the location of the individual's first job (which is an ancestor of the protected attribute), then ECO-fairness is not equivalent to the conditional demographic parity given all remaining attributes. However, suppose the unprotected attributes only include the descendants of the protected attribute, e.g. the individual's monthly expenditure. Then the ECO-fairness is equivalent to conditional demographic parity.

**The counterfactual fairness criterion.** The counterfactual fairness (cf) criterion was introduced by Kusner et al. (2017). A decision maker that targets cf corrects for historical disadvantages that may stem from protected attributes. Female applicants may have fewer opportunities for test preparation, leading to lower test scores and a lower likelihood of admission. Had they been male, they may have had more opportunities for preparation, achieved a higher score, and been admitted.

One of the goals of cf is to ensure that, all else being equal (such as effort or aptitude), an applicant would receive the same decision had they not been in a disadvantaged group. This goal is different from that of eco because it accounts for how the protected attribute affects the other attributes, how an applicant's gender affects their test score. Moreover, eco prohibits direct impacts of protected attributes on decisions but allows an indirect influence through unprotected attributes. In contrast, cf prohibits both types of influences, direct or indirect.

Mathematically, counterfactuals help capture this idea. Consider an applicant with test score $s$ and gender $a$. Had they belonged to a different group $a'$, they may have achieved a different test score $S(a') \,|\, \{A = a, S = s\}$. Notice this counterfactual score is conditional on the attributes of the applicant, including their (factual) score. The variable $S(\text{male}) \,|\, \{A = \text{female}, S = \text{high score}\}$ can capture that a female applicant with a high test score will have an even higher test score had she been male.

The cf criterion further uses a nested counterfactual. The variable $Y(a', S(a')) \,|\, \{A = a, S = s\}$ involves an intervention on the protected attribute along with a resulting change in the unprotected attribute. An cf-fair decision is one where this counterfactual does not change the distribution of the decision.

**Definition 2** (Counterfactual fairness (Kusner et al., 2017)). *A decision $Y$ exercises counterfactual fairness (cf) if, for all possible values of $a$, $a'$ and $s$,*

$$Y(a, S(a)) \,|\, \{A = a, S = s\} \stackrel{d}{=} Y(a', S(a')) \,|\, \{A = a, S = s\}.$$

This criterion requires that, for any individual with protected attributes $A = a$ and unprotected attributes $S = s$, their counterfactual decisions $Y(a', S(a'))$ must have the same distribution as if they had a different protected attribute value $A = a'$, together with the corresponding counterfactual unprotected attribute under this different protected attribute $S(a')$.

Algorithmic decisions that satisfy cf are not affected by (hypothetical) interventions on protected attributes, including accounting for how those interventions might subsequently change the other attributes. They use the protected attribute to correct for historical disadvantages.

## 3 Constructing fair algorithmic decisions

Defs. 1 and 2 provide criteria for fairness in terms of distributions of counterfactual decisions. Using the causal model of the decision-making process, these criteria help evaluate the fairness of its outcomes. The criteria can be used to evaluate decisions produced by a human process, such as an admissions committee, or produced by an algorithm, such as a fitted ml model.

Denote an algorithmic decision as $\hat{Y}$, a causal variable that depends on the attributes $\{a, s\}$. It comes from a fitted probabilistic decision maker, $\hat{Y} \sim f(a, s)$, where $f(\cdot)$ is a probability density function. For example, a binary admissions decision $\hat{Y} \sim f(a, s)$ is drawn from a Bernoulli that depends on the attributes (e.g., a logistic regression). By considering algorithmic decisions as variables in the causal model, we can infer algorithmic counterfactuals $\hat{Y}(a, s)$ and confirm whether they satisfy the eco and cf criteria.

Now consider a classical ml model that is fit to emulate historical admissions data. If the historical decisions did not exercise eco or cf then neither will the decisions produced by ml. Below we develop fair ml decision makers, algorithms that adjust the decisions made by classical ml to be eco-fair or cf-fair, i.e., to satisfy the eco and cf criteria. We will show that these adjusted decisions are as accurate as possible relative to the historical data, while still producing eco-fair or cf-fair decisions.

**Machine learning (ML) decisions.** An machine learning (ml) decision maker uses historical data to accurately predict the decision $Y$ from the protected attribute $A$ and unprotected attribute $S$. Admissions involves binary decisions, and so logistic regression is a common choice. The decision maker $f_{\text{ml}}(a, s)$ draws the decision from a Bernoulli, $\hat{Y}^{\text{ml}} \,|\, \{A = a, S = s\} \sim \text{Bern}(\sigma(\beta_S \cdot s + \beta_A \cdot a + \beta_0))$, where $\sigma(\cdot)$ is the logistic function and the coefficients are fit to maximize the observed data likelihood. When included in the causal model, $f_{\text{ml}}$ and $\hat{Y}^{\text{ml}}$ are illustrated in Fig. 2.

The ml decision $\hat{Y}^{\text{ml}}$ will accurately mimic the historical data. But it will also replicate harmful discriminatory practices. If the committee did not give equal opportunities to applicants of different genders then the

decision $\hat{Y}^{\mathrm{ml}}$ will violate eco. If the committee did not correct for the disparate impact of gender on the test score, then $\hat{Y}^{\mathrm{ml}}$ will not exercise cf.

Return to the illustrative simulation (Fig. 1). We fit a logistic regression to the training data, which finds coefficients close to the mechanism that generated the data. Consequently, when used to form algorithmic decisions, it replicates the unfair committee. Consider female applicant A ($a = \mathrm{f}, s = 85$) and male applicant B ($a = \mathrm{m}, s = 85$) and the classical ml decisions for each. For A, her probability of being admitted is 67%. For B, his probability of admission is 84%. Despite identical scores, the female applicant is 17% less likely to get in.

**Algorithmic decisions that satisfy eco.** We use the ml decision maker $f_{\mathrm{ml}}$ to produce an eco-fair decision maker $f_{\mathrm{eco}}$, one whose decisions satisfy eco. Consider an ml decision maker $f_{\mathrm{ml}}(a, s)$ and an applicant with attributes $\{a_{\mathrm{new}}, s_{\mathrm{new}}\}$. Her eco decision is $\hat{Y}^{\mathrm{eco}}(a_{\mathrm{new}}, s_{\mathrm{new}}) \sim f_{\mathrm{eco}}(s_{\mathrm{new}})$, where

$$f_{\mathrm{eco}}(s_{\mathrm{new}}) = \int f_{\mathrm{ml}}(a, s_{\mathrm{new}})p(a)\,\mathrm{d}a. \tag{1}$$

The eco decision probability holds the unprotected attribute $s_{\mathrm{new}}$ fixed and takes a weighted average of the ml decision maker for the different values of the protected attribute $a$. The weights are determined by the proportions of each group in the whole population. Note that the ml decision maker is fixed; it does not need to be retrained.

Return to Fig. 1. We use the fitted logistic regression $f_{\mathrm{ml}}(a, s)$ to produce the eco-fair decision maker. This data has equal numbers of women and men so the weighted average is $f_{\mathrm{eco}}(a, s) = 0.5 f_{\mathrm{ml}}(\mathrm{male}, s) + 0.5 f_{\mathrm{ml}}(\mathrm{female}, s)$. Using eco-fair decisions, applicants A and B both have a 77% probability of admission.

The eco decision $\hat{Y}^{\mathrm{eco}} \sim f_{\mathrm{eco}}(s_{\mathrm{new}})$ satisfies the eco criterion: applicants with the same score will have the same chance of admissions regardless of their gender. The intuition is that the eco decision preserves the causal relationship between the test score $S$ and the ml decision $\hat{Y}^{\mathrm{ml}}$, but it ignores the possible effect of the protected attribute $A$. In the *do* notation for interventions (Pearl, 2009), what this means is that $P(\hat{Y}^{\mathrm{eco}}; \mathrm{do}(a, s)) = P(\hat{Y}^{\mathrm{ml}}; \mathrm{do}(s))$ for all $a$. The weighted average is the adjustment formula (Pearl, 2009), which calculates $P(\hat{Y}^{\mathrm{ml}}; \mathrm{do}(s))$.

Finally, Thm. 1 shows the theoretical optimality of the eco decision makers among all those that satisfy eco.

**Theorem 1** (eco-fairness and optimality of eco decisions). *eco decisions satisfy the follow properties:*

 1. *The decision $\hat{Y}^{\mathrm{eco}} \sim f_{\mathrm{eco}}(a, s)$ is eco-fair.*

 2. *Among all eco-fair decisions, $\hat{Y}^{\mathrm{ECO}}$ maximally recovers the ML decision $\hat{Y}^{\mathrm{ml}}$,*

$$\hat{Y}^{\mathrm{ECO}} = \operatorname*{arg\,min}_{Y^{\mathrm{ECO}} \in \mathcal{Y}^{ECO}} \mathbb{E}\left[\mathrm{KL}(P(\hat{Y}^{\mathrm{ml}}(A, S)) || P(Y^{\mathrm{eco}}(A, S)))\right],$$

 *where $\mathcal{Y}^{ECO}$ is the set of eco-fair decisions and the expectation is taken over $P(A)P(S)$. (Proof in App. A.)*

Thm. 1 shows that $\hat{Y}^{\mathrm{ECO}}$ is the eco-fair decision maker that is closest to the ML decision maker in KL divergence, over a population where the protected and unprotected attributes are independent.

Thm. 1 focuses on probabilistic decision makers $\hat{Y}^{\mathrm{ml}}$, even though the optimal classifier that minimizes the misclassification rate is the deterministic Bayes classifier, which output decisions that are a deterministic function of the protected and unprotected attributes. The reason is that the ground truth outcomes are uncertain, e.g. an outcome of whether a student will be admitted to an Ivy League school entails randomness in the process. Given probabilistic ground truth outcomes, the optimal probabilistic decision maker $\hat{Y}^{\mathrm{ml}}$ minimizes the KL divergence to the ground truth outcomes, which is equivalent to minimizing the commonly used cross-entropy loss in a binary classification setting. While probabilistic decision makers $\hat{Y}^{\mathrm{ml}}$ may not be as accurate with respect to the misclassification rate as deterministic decision makers, they can reflect the uncertainty in the predictions, remaining faithful to the uncertainty in ground truth outcomes. In contrast, a

deterministic classifier ignores this uncertainty. We thus consider probabilistic decision makers for both ML and fair decision makers.

A reader may ask: in Thm. 1, why do we define optimality in terms of the KL distance between fair decision makers and ML decision makers, given that probabilistic ML decision makers could be suboptimal in classification? Why not minimize the distance to the ground truth outcomes? As it turns out, minimizing a decision maker's KL divergence to the optimal probabilistic decision maker $\hat{Y}^{\text{ml}}$ will also minimize its KL divergence to the ground truth outcomes (if the class of probabilistic ML decision makers is flexible enough). The reason is that the optimal probabilistic ML decision maker (in terms of KL divergence to the ground truth outcomes) must have the same distribution as the ground truth outcomes,

$$\hat{Y}^{\text{ml}}(a,s) = \arg\min_{\tilde{Y}} \text{KL}(P(Y(a,s))||P(\tilde{Y}(a,s))) \quad \forall a,s \qquad \Leftrightarrow \qquad \hat{Y}^{\text{ml}}(a,s) \stackrel{d}{=} Y(a,s) \quad \forall a,s.$$

Thus, considering the KL divergence to optimal probabilistic decision makers makes a meaningful metric; a decision maker that minimizes this distance implicitly minimizes the KL divergence to the (random) ground truth outcomes.

**FTU, $\hat{Y}^{eco}$, and why consider the population of $P(A)P(S)$ in Thm. 1.** An alternative method to satisfy eco is ftu (Kusner et al., 2017). ftu satisfies eco by fitting an ml model from the unprotected attribute $S$ to the decision $Y$, and completely omitting the protected attribute $A$; ftu decisions are also *eco*-fair. While both ftu and the eco decision makers satisfy the eco criterion, ftu may still indirectly correlate with the protected attributes if an unprotected attribute correlates with the protected attribute in the population. In contrast, the eco decision maker would not.

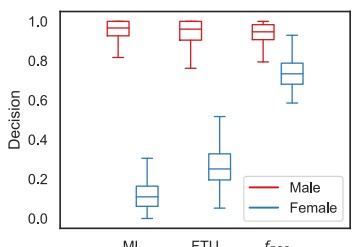

As an example, we simulate an additional unprotected attribute that has a 0.9 correlation with gender. We further subsample the adult dataset such that the correlation between the protected attribute and the decision is 0.7. We then compare the ftu decision and the eco. While both eco-fair, Fig. 3 shows that the ftu decisions remain highly correlated with gender while the eco decision has a substantially lower correlation. The eco decision avoids inheriting discrimination patterns when unprotected attributes are highly correlated with the protected attribute. Such a high correlation between protected and unprotected attributes do not appear in the adult dataset; the prediction accuracy of ftu and $f_{eco}$ are similar. That said, Fig. 3 delineates a setting where the eco decision maker may be desired.

**Figure 3:** fairness through unawareness (ftu) vs. $f_{eco}$ ; The eco decision avoids inheriting discrimination patterns when unprotected attributes are highly correlated with the protected attribute. While both eco fair, ftu decisions remain highly correlated with gender while the eco decision has a substantially lower correlation.

This distinction between ftu and the eco decision makers also suggests we consider the population of $P(A)P(S)$ in Thm. 1. The theorem shows that the eco decision maker is closest in expected KL to the ML decision maker, among all fair decision makers. Note the expectation is taken with respect to $P(A)P(S)$, a distribution where there is no information 'leaked' from the protected attribute to the unprotected attributes. The reason is that if there is correlation between them, then we could consider "fair" decision makers that are not actually fair – they might make decisions based on unprotected attributes but really be capitalizing on correlations to the protected ones. The optimality of the fair decision maker is under the setting where we only pick attributes that are uncorrelated to the protected one.

**Algorithmic decisions that exercise cf.** We now show how to adjust the eco-fair decision maker to satisfy counterfactual fairness.

Consider the eco-fair decision maker $f_{\text{eco}}(s)$ and an applicant with attributes $\{a_{\text{new}}, s_{\text{new}}\}$. Her cf-fair decision is $\hat{Y}^{\text{cf}} \sim f_{\text{cf}}(a_{\text{new}}, s_{\text{new}})$, where

$$f_{\text{cf}}(a_{\text{new}}, s_{\text{new}})) = \iint f_{\text{eco}}(s(a)) \, p(s(a) \,|\, a_{\text{new}}, s_{\text{new}}) p(a) \, \mathrm{d}s(a) \, \mathrm{d}a. \tag{2}$$

Thus we form an cf decision $\hat{Y}^{\text{cf}}$ by drawing from a mixture: (a) sample a gender from the population distribution $a \sim p(a)$; (b) sample a test score from its counterfactual distribution $s(a) \sim p(s(a) \,|\, a_{\text{new}}, s_{\text{new}})$;

(c) sample the eco-fair decision for that counterfactual test score $\hat{y}^{\mathrm{cf}} \sim f_{\mathrm{eco}}(s(a))$. We repeat this sampling over different values of $a$ and $s(a)$ to form the cf decision $\hat{Y}^{\mathrm{cf}}$.

One subtlety of $\hat{Y}^{\mathrm{cf}}$ is step (b). It draws the counterfactual test score under an intervened gender, but conditions on the observed gender and test score. This requires *abduction* (Pearl, 2009), where unobserved noise variables are inferred conditioned on observed variables, allowing us to sample counterfactual outcomes and construct CF-fair decision makers. For example, consider a model where females have fewer opportunities for test preparation. If we observe a female applicant with a high test score, the unobserved noise might capture that this applicant is particularly gifted. Under this value of noise, her counterfactual test score will be even higher.

Put differently, the cf-fair decision maker uses the eco-fair decision maker $f_{\mathrm{eco}}(s)$, which only depends on the test score (and averages over the gender). However, the cf-fair decision maker also corrects for the effect on the test score due to the gender of the applicant. It replaces the current test score $s_{\mathrm{new}}$ with the adjusted score $s(a) \mid \{a_{\mathrm{new}}, s_{\mathrm{new}}\}$ under $a \sim p(a)$. With this corrected score, it produces an cf-fair decision.

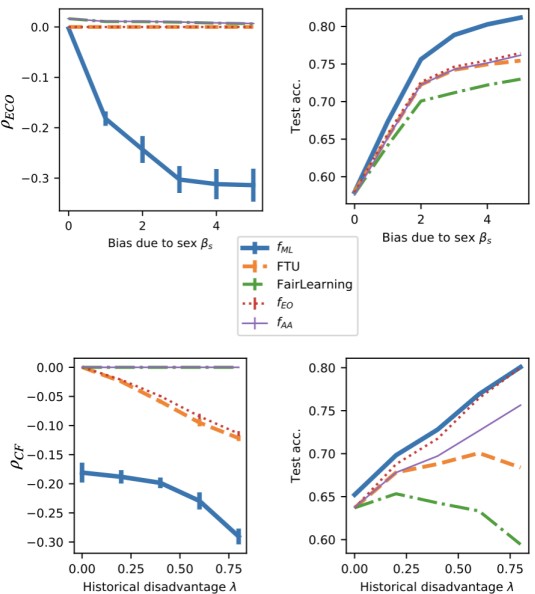

For the simulated data in Fig. 1, we calculate $f_{\mathrm{cf}}(a, s)$ from $f_{\mathrm{eco}}(a, s)$. This decision maker requires abduction, calculating the test score each applicant would have achieved had their gender been different. Consider applicant C. Her eco-fair probability of acceptance is 69%, but when exercising cf it increases to 70%. This adjustment corrects for the (simulated) systemic difficulty of females to receive test preparation and, consequently, higher scores.

The cf-fair decision maker $f_{\mathrm{cf}}(a, s)$ accounts for historical disadvantage because it adjusts the applicant's test score (and their resulting admissions decision) to their counterfactual test scores under intervention on gender. Further, each element is computable from the dataset. (We discuss how to calculate these decisions below.) Note that cf-fair decisions can be formed from any eco-fair decision maker.

**Figure 4:** Measures of decision-quality in the simulated admissions data (error bars indicate $\pm$ 1 sd). The eco decision maker $f_{\mathrm{ECO}}$ is the most accurate one that achieves eco-fairness; the cf decision maker $f_{\mathrm{cf}}$ is the most accurate one that satisfies cf-fairness. The classical ML decision maker $f_{\mathrm{ml}}$ is most accurate overall but violates both fairness measures. We vary the historical disadvantage $\beta_a$ from 0.0 to $+5.0$ and vary direct gender discrimination $\lambda$ from 0.0 to $+0.8$.

The cf-fair decision maker depends on the protected attribute, and thus the cf-fair decision $\hat{Y}^{\mathrm{cf}}$ is not eco-fair. But among all cf-fair decisions, it is closest to the eco-fair decision; we prove this fact in Thm. 2.

**Theorem 2** (cf-fairness and optimality of the cf decisions ). *cf decisions satisfy the following properties:*

*1. The decision $\hat{Y}^{\mathrm{cf}} \sim f_{\mathrm{cf}}(a, s)$ is cf-fair.*

*2. Among all cf decisions, the cf decision maker minimally modifies the marginal distribution of $Y^{\mathrm{eco}}$,*

$$\hat{Y}^{\mathrm{cf}} = \underset{Y^{\mathrm{cf}} \in \mathcal{Y}^{\mathrm{cf}}}{\arg\min} \, \mathbb{E}\left[\mathrm{KL}(P(\hat{Y}^{\mathrm{eco}}(A, S)) || P(Y^{\mathrm{cf}}(A, S)))\right],$$

*where $\mathcal{Y}^{\mathrm{cf}}$ are all cf decisions and the expectation is over $P(A)P(S)$. It also preserves the marginal distribution of the eco decision maker, $P(\hat{Y}^{\mathrm{cf}}) = P(\hat{Y}^{\mathrm{eco}})$. (Proof in App. B.)*

Thm. 2 also says that cf-fair decisions preserve the marginal distribution of the eco-fair decisions. If the eco decision maker admits 20% of the applicants then the cf decision maker will also admit 20%. (The cf

decision maker will likely admit a different set of applicants from the eco decision maker.) That it preserves the marginal distribution makes the cf-fair decision maker applicable as a decision policy, such as when there is a fixed budget for admissions.

Finally, cf-fair decisions also satisfy demographic parity, a group-level statistical criteria that has been used as a measure of cf (Dwork et al., 2012). Demographic parity requires equal decision distributions for the advantaged and disadvantaged groups of the protected attribute. (It does not involve a causal model.) Kusner et al. (2017) shows that, assuming the causal model in Fig. 2, decisions that satisfy counterfactual cf also satisfy demographic parity.

**Why are the eco and cf decisions optimal?** Thms. 1 and 2 establish the optimality of the the eco and cf decisions; they rely on the following observation. The potential for unfairness arises when individuals have different values in their protected attributes. In the admissions example, applicants have different genders (protected attribute), which may causally affect their admission decisions (outcome) and lead to unfairness. If all applicants had the same gender then any decision maker would be "fair" in that there is no possibility for discrimination.

This observation leads to the theorems. The eco and cf decision makers predict in a fictitious world where all applicants had the same gender $A'$. They take an existing decision maker and ask: what would its counterfactual prediction be if each applicant had the gender $A'$? The eco decision (Eq. 1) averages over a distribution $P(A')$. Thm. 1 says that the eco decisions are closest to the ML decisions when $A'$ is distributed as the gender distribution in the data. Thm. 2 says that the cf decision maker is closest to (i.e. "minimally modifies") the eco decision maker.

The decisions are probabilistic, so closeness in the two theorems is measured by the average KL distance. The average is over a target population where the protected $A$ and unprotected $S$ are independent; this mimics an ideal setting that fair learning algorithms target, i.e. where no applicants are disadvantaged.

**Calculating eco and cf decision makers.** The eco and cf decision makers can be calculated from data. The eco decision maker uses the fitted ML decision maker $f_{\mathrm{ml}}$, but marginalizes out the protected attribute. The cf decision maker uses the eco decision maker $f_{\mathrm{eco}}$, including an abduction step (Pearl, 2009) to calculate the counterfactual value of the attribute $s(a')$.

We focus on settings where the counterfactual decisions are identifiable from observational data. We assume: (1) The protected attributes $A$ and unprotected attributes $S$ follow the causal graph in Fig. 2. That is, there is no unobserved confounding between $A$ and $S$, between $S$ and $Y$, and between $A$ and $Y$. (2) The structural equation model for $A, S, Y$ needs to be correct so that structural equation can be identified from observational data. These two conditions enable the identification of counterfactual decisions.

Given the necessary assumptions for identifiability, the counterfactual calculation requires modeling the effect of the protected attribute $A$ (gender) on the other attribute $S$ (test score). With linear models, this calculation involves combining the residual on the observed data with a prediction about the coun-

---

**Algorithm 1** The eco and cf decision makers (for additive-error models).

**Input:** Data $\mathcal{D} = \{(a_i, s_i, y_i)\}_{i=1}^n$, where $a_i$ is protected, $s_i$ is not, and $y_i$ is the decision.

**Output:** Decision makers $\{f_{\mathrm{ml}}(a, s), f_{\mathrm{eco}}(s), f_{\mathrm{cf}}(a, s)\}$

From the data $\mathcal{D}$, fit $f_{\mathrm{ml}}(a, s)$, $p(a)$, and $g(a) = \mathbb{E}\left[S \mid A = a\right]$ (e.g., with regression).

①   The ml decision maker $f_{\mathrm{ml}}(a, s)$ draws from

$$\hat{y}^{\mathrm{ml}} \sim f_{\mathrm{ml}}(a, s).$$

②   The eco decision maker $f_{\mathrm{eco}}(s)$ draws from Eq. 1,

$$a' \sim p(a); \qquad \hat{y}^{\mathrm{eco}} \sim f_{\mathrm{ml}}(a', s).$$

③   The cf decision maker $f_{\mathrm{cf}}(a, s)$ draws from Eq. 2,

$$a' \sim p(a); \quad s' = g(a') + (s - g(a)); \quad \hat{y}^{\mathrm{cf}} \sim f_{\mathrm{eco}}(a', s').$$

---

terfactual setting. In the data of Fig. 1, the abduction infers what applicant C's test would have been had she been male, given her test score (as a female) was 65. The algorithm can also be applied to nonlinear models; see App. C for details.

As a concrete example, Alg. 1 provides the algorithm for calculating eco and cf decision makers (Eqs. 1 and 2) from a fitted ml decision maker in the case of additive-error models, e.g. $s = g(a) + \epsilon$ for some function $g$ and some random variable $\epsilon$. (We discuss the identification of the eco and cf decision makers and prove the correctness of Alg. 1 in App. C.)

The FairLearning (Kusner et al., 2017) algorithm, which also satisfies cf-fairness performs the same abduction step as $f_{\mathrm{cf}}$ to compute residuals $\varepsilon_i = s_i - \mathbb{E}[S \mid A = a_i]$ but then fits a decision maker using only the residuals and non-descendants of the protected attributes. For example, in admissions, after calculating the residual it does not include the test score in its decision maker. In contrast, Alg. 1 uses all available attributes and still is cf-fair. The eco decision maker in Alg. 1 also makes use of all attributes; it does not omit protected attributes as done in ftu (Kusner et al., 2017). Empirically, Alg. 1 provides more accurate decisions than these existing eco and cf algorithms.

**Multiple protected and unprotected attributes.** Alg. 1 can be generalized to settings with multiple protected and unprotected attributes. Note that both $A$ and $S$ can be a vector, so we can collect all protected attributes into the $A$ variable and all unprotected attributes into the $S$ variable. As long as the resulting attributes still follow the causal graph in Fig. 2, Alg. 1 is still correct. That said, Alg. 1 may not apply in other cases, e.g. when the following three conditions simultaneously hold: (1) the multiple unprotected attributes are chained; (2) we consider interventions on only a subset of the unprotected attributes; (3) we focus on a subpopulation by conditioning on a descendant of the intervened unprotected attributes. In this case, the corresponding counterfactual is not identifiable.

## 4 Empirical studies

We study algorithmic decisions on both simulated and real datasets, examining the tradeoff between fairness and prediction quality. We compare the eco and cf decision-makers to existing algorithmic decision-makers—fairness through unawareness (ftu) and FairLearning (Kusner et al., 2017)—that target the same eco and cf fairness criteria. Throughout the empirical studies, we consider ML decision-makers that are (generalized) linear models. We use linear regression models when the outcome is real-valued, and logistic regression when the outcome is binary. (The supplement provides software that reproduces the studies.)

We find the following. (1) As expected, the classical ml decision $\hat{Y}^{\mathrm{ml}}$ is accurate but unfair. (2) The eco decision $\hat{Y}^{\mathrm{eco}}$ is less accurate than $\hat{Y}^{\mathrm{ml}}$, but is eco-fair; it is more accurate than ftu. (3) The cf decision $\hat{Y}^{\mathrm{cf}}$ is less accurate than the eco-fair decision, but is cf-fair and achieves demographic parity; it is more accurate than FairLearning.

**Evaluating algorithmic decision-makers.** For each decision-maker, we measure the eco and cf fairness of its decisions, and evaluate its fidelity relative to the historical data.

One way to measure fairness is to compare distributions of counterfactual decisions. Suppose a protected attribute $A$ with values $a$ (for the advantaged group) and $a'$ (for the disadvantaged group). As one metric, we consider the average violations of eco and cf over the population as metrics: $\rho_{\mathrm{eco}} = \frac{1}{n} \sum_{i=1}^{n} \left[ \mathbb{E}\left[ \hat{Y}(a, s_i) \mid a_i, s_i \right] - \mathbb{E}\left[ \hat{Y}(a', s_i) \mid a_i, s_i \right] \right], \rho_{\mathrm{cf}} = \frac{1}{n} \sum_{i=1}^{n} \left[ \mathbb{E}\left[ \hat{Y}(a, s(a)) \mid a_i, s_i \right] - \mathbb{E}\left[ \hat{Y}(a', s(a')) \mid a_i, s_i \right] \right]$. The sum is over $n$ samples in a held-out test set; expectations are with respect to the decision-maker.

The eco metric contrasts the decision probabilities when changing the protected attribute but holding other attributes fixed. The cf metric contrasts the decision probabilities when changing the protected attribute $A$ and allowed other attributes to counterfactually vary. When the eco metric is 0.0, the decision-maker achieves eco-fairness; when it is greater than 0.0, there is bias towards the disadvantaged group; when it is less than 0.0, the decision-maker favors the disadvantaged group. The same interpretation applies to cf.

Another measure of cf fairness is demographic parity Dwork et al. (2012). We measure demographic parity with the symmetric Kullback-Leibler (kl) divergence between prediction distributions for the decision $P(\hat{Y} \mid A = a)$ and $P(\hat{Y} \mid A = a')$; it is zero for a decision-maker that achieves demographic parity. (We evaluate the symmetric kl by binning the values of predictions.)

Finally, we measure the fidelity of the decision-maker to the historical data on which it was fit. The metric is the prediction score: for binary outcomes, it is the mean accuracy of out-of-sample predictions; for real-valued outcomes, it is the coefficient of determination $R^2$ of the predictions.

**Simulated admissions.** We first study simulated datasets about an unfair admissions committee, which is from the following structural model, $a_i \sim \mathrm{Bernoulli}(0.5); s_i \mid a_i = \max(0, \min(\lambda \cdot a_i + 100 \cdot \varepsilon, 100)); \varepsilon \sim \mathrm{Uniform}[0,1]; y_i \mid a_i, s_i \sim \mathrm{Bernoulli}(\sigma(-1.0 + \beta_s \cdot s + \beta_a \cdot a))$.

(We threshold the test score to mimic real-world test scores that are usually bounded.) We fix the effect of test score on admissions to $\beta_s = 2.0$. We generate multiple datasets by varying the gender bias $\beta_a$ and the historical disadvantage on test score $\lambda$.

Fig. 4 show how eco-fairness and prediction quality trade off as the bias $\beta_a$ increases. Only the eco decision-makers $f_{\mathrm{ECO}}$ and ftu achieve eco fairness. Although both decision-makers are less accurate than classical ml, the eco decision-maker is more accurate than ftu. Fig. 4 shows how cf-fairness and prediction quality trade off as the historical disadvantage $\lambda$ increases. Only the $f_{\mathrm{cf}}$ and FairLearning decision-makers achieve cf-fairness. Among these cf-fair decision-makers, the cf decision-maker of this paper is more accurate.

**Case studies.** We study eco and cf decision-makers on a simulated admissions dataset and three real datasets of sensitive decisions about people. The adult income data (Dua & Graff, 2017a) and the German credit data (Dua & Graff, 2017b) contain data about people and decisions about which are loan worthy.[1]

| | Metrics ($\times 10^2$) on Adult | | | |
|---|---|---|---|---|
| | $\rho_{\mathrm{eco}}$ | $\rho_{\mathrm{cf}}$ | **KL** | **Prediction** |
| $f_{\mathrm{ml}}$ | 2.5(1.9) | 16.2(12.0) | 15.0 | 78.6 |
| ftu | **0(0)** | 14.8(7.9) | 12.2 | 77.3 |
| $f_{\mathrm{eco}}$ | **0(0)** | 14.1(8.3) | 12.6 | **77.4** |
| FL | -14.8(9.1) | **0(0)** | 5.3 | 75.1 |
| $f_{\mathrm{cf}}$ | -9.1(9.8) | **0(0)** | **1.5** | **77.1** |

**Figure 5:** In the Adult dataset, algorithmic decision-makers decide which individuals are loanworthy based on their income and we examine their fairness relative to gender. The eco and cf decision makers are fair towards females while remaining the most accurate. Both the eco decision maker $f_{\mathrm{eco}}$ and ftu achieve eco-fairness as measured by the eco-metric. The cf decision-maker $f_{\mathrm{cf}}$ and FairLearning (FL) (Kusner et al., 2017) are cf-fair and achieve demographic parity (close-to-zero KL). The ml decision maker $f_{\mathrm{ml}}$ is the most accurate overall but the eco decision maker is the most accurate among the eco-fair decision-makers. We report mean values across individuals with the standard deviation in parentheses. eco and cf metric standard deviations are $\leq 0.1$ and $\leq 0.11$, respectively. We also report the KL divergence and prediction accuracy, which are distributional metrics.

ProPublica's COMPAS data contains information about criminal defendants and decisions about their recidivism score. Each dataset contains protected and unprotected attributes, and in the studies we adjust classical ml decision-makers to achieve fairness relative to the protected attributes. We focus on the adult income data in this section and defer the results of the German credit data and the COMPAS data to App. F.

Fig. 5 summarize the fairness and prediction quality from the adult income data. In the adult income data, race and gender are the protected attributes. We discuss the fairness metrics $\rho_{eco}$ and $\rho_{cf}$ relative to gender and report the mean (and s.d.) of all the metrics. The findings are consistent with the simulation studies and generalize across the studies.

Although the classical ml decision-maker is the most accurate, its decisions are biased against the disadvantaged group. Positive values in the EO-metric $\rho_{eco}$ reveal direct discrimination in classical ml decisions; males and white individuals receive a higher probability of being decided as loan-worthy or lower recidivism score than their equivalent female or non-white counterparts. Positive values in the AA-metric $\rho_{cf}$ indicate that the classical ml decision-maker does not exercise cf. It provides lower decision probabilities to females or non-white individuals than their male or white counterparts whose attribute values also reflect the historical advantage.

---

[1] Readers who are familiar with these datasets know that they contain scores, as opposed to decisions. We use these scores to create "decisions" in order to evaluate the methods of this paper on binary decisions.

The eco and ftu decisions are eco-fair ($\rho_{eco}$ is zero) and the eco decisions are more accurate. The eco-fair decision-makers provide equal decision probabilities to individuals that are equal in all other attributes, regardless of gender or race. Neither counterfactual cf nor FairLearning achieve eco-fairness, but this is by design. These decisions produce negative values for the eco metric, showing an advantage for the disadvantaged group. The cf-fair decision-makers correct for historical biases, which leads to different decisions for the disadvantaged and advantaged groups.

The cf and FairLearning decisions both achieve cf-fairness ($\rho_{cf}$ is zero) and demographic parity (kl is zero). They succeed in providing equal decision probabilities to females or non-white individuals after considering their counterfactual attribute values had they been male or white. Among these cf-fair decision-makers, $f_{cf}$ is more accurate.

**Measuring decisions in the adult income data.** We study the algorithmic decisions in more depth. Using the adult income data, we examine the metrics of Fig. 5. (The results below exhibit similar patterns in all three datasets.)

We first discuss eco-fairness for gender. Fig. 6a reveals the same patterns of eco-fairness that Fig. 5 showed. Both eco and ftu decision probabilities align with the diagonal; they are eco-fair in providing equal decision probabilities to equally qualified individuals regardless of gender. None of the other algorithmic decision makers is eco-fair. Fig. 6a shows that classical ml decisions are biased against females, producing lower decision probabilities for them than their male counterparts. In contrast, the cf-fair decision makers $f_{cf}$ and FairLearning are less eco-fair than $f_{ml}$ but provide an advantage to females after accounting for historical disadvantages. Further, cf and FairLearning are less eco-fair than classical ML.

We next discuss cf-fairness for gender. Fig. 6b compares the decision probabilities received by individuals if they were females to those they receive if they were males, with the resulting changes to their other attributes. An algorithmic decision maker is cf-fair if the decision probabilities align with the diagonal. The decision makers $f_{cf}$ and FairLearning produce decision probabilities that align with the diagonal and are thus cf fair. The remaining decision makers do not exercise cf; eco decisions provide a greater degree of cf-fairness than ftu decisions while classical ml decisions are the most unfair in exercising cf.

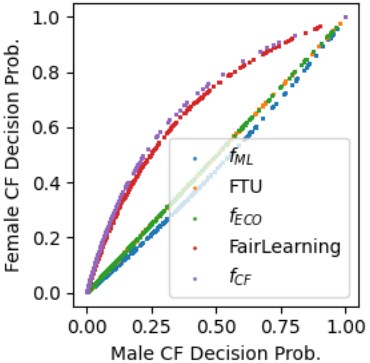

**(a)** Assessing eco

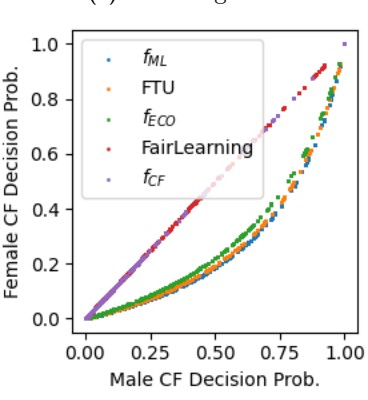

**(b)** Assessing cf

**Figure 6:** Comparing individual decision probabilities under (hypothetical) "intervention" on gender (in the adult income data). (a) To study eco, the other attributes remain at their original values and we plot $p(\hat{y}(a,s) \,|\, a,s)$ against $p(\hat{y}(a',s) \,|\, a,s)$. The $f_{\mathrm{ECO}}$ decision maker provides eco to females; it produces decision probabilities that align along the diagonal. (b) To study cf, the remaining attributes vary with the intervention on gender and we plot $p(\hat{y}(a,s) \,|\, a,s)$ against $p(\hat{y}(a',s(a')) \,|\, a,s)$. The decision maker $f_{\mathrm{cf}}$ exercises cf; it provides equal decision probabilities to females as their male counterparts after correcting for the historical disadvantage they face. In contrast, the classical ml decision maker $f_{\mathrm{ml}}$ demonstrates both direct and indirect discrimination against females; it provides lower decision probabilities to females compared to their male counterparts in both comparisons.

We next inspect demographic parity. Fig. 7 illustrates demographic parity by plotting the decision probabilities for both male and female individuals. The left panel comes from a classical ml decision maker; the right panel

comes from $f_{cf}$. While the classical ml decision maker produces noticeably different decision probabilities for the two groups, the decision probabilities produced by the cf decision maker $f_{cf}$ are nearly identical.

We now turn to the predictive performance. Fig. 5 reports the predictive performance of all methods. Though it is not fair by either criterion, classical ml produces the "best" predictions, closest to the historical dataset. Among the eco-fair methods, eco produces better predictions than ftu. This corroborates Thm. 1. Moreover, eco's prediction scores are close to classical ml, so its fairness comes with little cost. Among the cf-fair methods, cf predicts better than FairLearning; this corroborates Thm. 2. But since it alters the decision probability of individuals from disadvantaged groups, it predicts less well than the ml methods.

## 5    Discussion

We develop fair ML algorithms that modify fitted ML predictors to make them fair. We prove that the resulting predictors are eco-fair or cf-fair, and they otherwise maximally recover the fitted ML predictor.

There are a few limitations of this work. One limitation is in the scope of fairness notions being studied. We focus on two counterfactual notions of fairness in this work; the proposed algorithms may only be fair with respect to these notions; they may not improve fairness with respect to other fairness notions. Develop post-hoc algorithms that simultaneously enforce multiple fairness notions is an interesting direction.

Another limitation lies in the assumptions required by counterfactual approaches to fairness. The methods in this paper rely on a correct causal model, the ability to estimate its parameters, and the ability to estimate the necessary counterfactuals. We recognize that these are strong requirements—positing

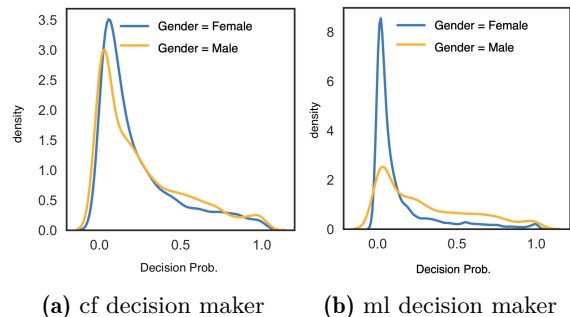

**(a)** cf decision maker    **(b)** ml decision maker

**Figure 7:** The adult income data: distributions of decision probabilities. (a) The cf-fair decision maker $f_{cf}$ produces equal decision probabilities for females and males, achieving demographic parity with respect to gender. (b) The classical ml decision maker $f_{ml}$ does not satisfy demographic parity.

and fitting causal models requires both domain expertise and statistical care. Moreover, even with the correct model in hand, issues of unobserved confounding can affect the ability to properly estimate its parameters or the counterfactuals. Developing fairness algorithms robust to violations of these assumptions is another worthwhile avenue of future work.

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
