# OpenReview forum: "Adjusting Machine Learning Decisions for Equal Opportunity and Counterfactual Fairness"
_TMLR — Accepted by TMLR_

### Review · Reviewer_qyZv · 2022-10-22

**Summary Of Contributions:**

This paper develops methods to post-process a prediction function to achieve two causal fairness criteria: "equal counterfactual opportunity'' and "counterfactual fairness.''

The authors consider the following setting: let $s_i$ denote the protected attribute, $a_i$ denote the unprotected attributes, and $y_i$ denote the decision. For example, $s_i$ may be gender, $a_i$ may be test scores, and $y_i$ may be an admissions decision made by a committee. The authors assume that the setting is summarized by a causal model in which (i) both $s_i, a_i$ affect $y_i$; and (ii) $s_i$ may affect $a_i$ (Figure 2). Let $Y_i(s, a)$ denote the counterfactual/potential decision and $A(s)$ denote the counterfactual/potential characteristics.

In this setting, the authors define:
* Equal counterfactual opportunity (ECO): the decision $Y$ satisfies ECO if $Y(s, a) | \{S = s, A = a\} \sim Y(s', a) \mid \{S = s, A = a\}$. In words, group membership has no direct causal effect on the decision among similarly qualified individuals.
* Counterfactual fairness (CF): the decision $Y$ satisfies CF if $Y(s, A(s)) \mid \{S = s, A = a\} \sim Y(s', A(s')) \mid \{S = s, A = a\}$. In words, group membership has no effect on the decision accounting for the possible causal effects it has on the other unprotected attributes, effectively "correcting'' for historical disadvantages/disparities that may be reflected in the unprotected attributes.

The authors next propose post-processing techniques to post-process an existing prediction function to satisfy ECO and CF:
* To satisfy ECO, the authors propose to construct a new prediction function that takes a weighted average of the existing prediction function across different values of the protected attribute at each $A = a$. The authors show that the resulting predictor $\hat{Y}^{ECO}$ provides the best approximation (in the sense of minimizing the KL-divergence between the new predictions and the existing predictions) to the existing prediction function among all that satisfy ECO (Theorem 1).
* To satisfy CF, the authors propose to construct a new prediction function that post-process an ECO-fair prediction function based on the sampling procedure described on pg. 7. The authors show that the resulting CF prediction function provides the best approximation (again in the KL-divergence minimizing sense) to the ECO-fair prediction function.

The authors illustrate their proposed methods in simulations, and applications to the adult income data, Compas data, and German credit data.

**Broader Impact Concerns:**

I have no concerns about the ethical implications of this work.

**Requested Changes:**

See my discussion of the weaknesses above.

My main concerns are (i), (ii) above. These concerns could be addressed by either (i) showing that the results extend to the case in which we evaluate KL-divergence over the underlying joint distribution or (ii) explicitly discussing the limitations of this result per my comments above.

Addressing the remaining weaknesses by adding some further discussion will strengthen the work.

**Strengths And Weaknesses:**

Strengths:
(i) The paper provides an easy-to-follow discussion of two reasonable causal fairness criteria (ECO and CF). It provides simple post-processing techniques to achieve these criteria, and provides extensive empirical studies on standard datasets to show their performance relative to existing methods in the causal fairness literature.

Weaknesses:
(i) The conditions for Theorem 1 are not useful -- in particular, it defines the expectation over $P(S) P(A)$ i.e., assuming that $S$ is independent of $A$. But, as the authors discuss at length in the beginning of the paper, we are often worried that there is a causal effect of $S$ on $A$ in which case $S$ is not independent of $A$. So, how is this then a useful guarantee if it does not apply to the actual data-generating process/causal graph at hand? The same weakness also applies to their construction of the CF-fair prediction function and its guarantee in Theorem 2.

(ii) Assuming for simplicity we care about squared-loss, the "best'' FTU prediction function is simply $E[Y \mid A]$ (i.e., the true conditional expectation of $Y$ given the unprotected attributes $A$). The "best'' unconstrained prediction function would simply be $E[Y \mid A, S]$, which we can think of as the ideal ML decision-maker to use the authors' language.
From this perspective, we can think of both the FTU prediction function and the authors' procedure as post-processing the ideal ML decision-maker. The FTU prediction function applies iterated expectations $\hat{Y}^{FTU}(a) = \int E[Y \mid A = a, S = s] P(s \mid a) ds$, and weights the ML decision maker according to the conditional distribution $p(s \mid a)$. In contrast, the ECO prediction function weights the ML decision maker according to the marginal distribution $p(s)$.
So these only differ by their choice of weights to apply to the ideal ML decision-maker.

Furthermore, based on the proof of Theorem 1 and this intuition, it seems like if we actually computed the KL-divergence under the true joint distribution $P(A, S)$, we may then find that the FTU prediction function actually provides the best approximation to the ideal ML learner. Is that conjecture correct? If not, it would be value to discuss. If so, then it is not clear why we should use the authors' post-processing algorithm in the first place. Similarly, since the fairlearning algorithm also satisfies CF-fairness, does the analogous result hold for it?

(iii) I found the authors' discussion of FTU to be unconvincing on pg. 7. In particular, the authors argue that FTU will satisfy to some other, undefined notion of fairness (i.e., "undesirable consequences if an unprotected attribute correlates..."). It is odd to criticize FTU for something that it doesn't claim to achieve (and something the authors don't even formally define).

(iv) Does the abduction step referred to on pg. 8 refer to the sampling procedure in the construction of the CF fair predictor? If so, that should be mentioned. Otherwise, the abduction step is not defined in the paper, and I was confused by what the authors were referring to.

---

### Review · Reviewer_FtKh · 2022-11-13

**Summary Of Contributions:**

The authors propose a method for post-hoc adjustment of machine learning predictions using counterfactual reasoning. In particular, the authors introduce equal counterfactual opportunity (ECO), a definition centered around equivalent decision distributions agnostic to the protected attribute s. The authors show that their decision makers that optimally modify the marginal distribution of the corresponding decision Y for ECO and counterfactual fairness (Kusner et al 2017). Lastly, they present empirical experiments using a college admissions example comparing their defined bias metric, KL divergence between the proposed decision distribution and the vanilla ML model distribution, and the overall model performance.

**Broader Impact Concerns:**

With the topic of counterfactual fairness, it is always important to specify the underlying assumptions. This paper does an excellent job of that. No other concerns about broader impact concerns.

**Requested Changes:**

See W1-W3

**Strengths And Weaknesses:**

Overall, I really enjoyed reading this paper.

S1) Counterfactual fairness is a tricky topic to cover correctly, and I found that the authors did a clear job of motivating the work, setting up the problem, and presenting meaningful results. One of the best parts of the paper is the clear engagement of the limitations of this work, for example the explicit need for a correct causal model of the world and no unobserved confounding.

S2) The experiments were clear and well-designed. Certainly followup work could explore cases where the proposed model might run into technical errors (e.g., finite sample issues), but the presented empirical work is compelling.

W1) The diferences in metrics on Figure 4 can be quite small, and it was unclear how to think about the effect of sample size. Confidence intervals or bootstrapping data splits would help clarify.

W2) The notation for Definition 1 and 2 is nonstandard, and I found it very difficult to parse the text and the notation without an explicit statement about what the Y(s,a) | {.} or A(s') | {.} notations meant. One additional line in the text would have cleared that up.

W3) Additionally, I spotted a few typos that are no terrible but should be amended:
 - p7, "is particularly gifted. under this value of noise" -> ". Under"
 - p7, "also corrects for the affect on the test score" -> "effect"

---

### Review · Reviewer_s9RH · 2022-11-28

**Summary Of Contributions:**

This paper proposes two methods for training fair classifiers (aka decision processes), one designed to satisfy a fairness criterion they call Equal Counterfactual Opportunity (ECO) and one designed to satisfy Counterfactual Fairness (CF) as proposed by Kusner et al. (2017).

Each method proceeds by first training an unconstrained probabilistic classifier and then applying a simple modification that gives rise to a new, fair probabilistic classifier. Each method is supported by a theorem which shows that the method is "optimal," in the sense that among all methods that satisfy the corresponding criterion, their method minimizes the KL divergence between the initial probabilistic classifier and the resulting fair classifier.

They compare their methods empirically on a simulated dataset and three real datasets, under causal assumptions sufficient to render identifiable the counterfactual quantities required for Counterfactual Fairness. They compare their ECO-fair method to Fairness Through Unawareness (FTU), which also trivially achieves ECO-fairness; and they compare their CF method to the original FairLearning method proposed by Kusner et al. (2017).

**Broader Impact Concerns:**

No concerns.

**Requested Changes:**

- **[critical]** Unless I have seriously misunderstood something, the proofs need to be checked/corrected, and then the results need to be re-run using the corrected method described above.
- **[critical]** I think a discussion is required about why the divergence property described in the theorems (which should hold with the corrected method) is desirable/meaningful. Or perhaps there are other desirable properties that these methods have that can be presented instead or in addition.
- **[critical]** In the empirical studies, what kind of structural model do they assume? Do they assume linear models throughout?
- **[critical]** Figure 5 is too small to read fully.
- **[would strengthen the work]** ECO-fairness is equivalent to conditional demographic (aka statistical) parity, conditional on all remaining features. It's worth acknowledging that while causal reasoning might plausibly lead to ECO-fairness, causal assumptions and causal inference techniques are not required for their ECO-fair method. This equivalence also suggests using other techniques as a baseline, since there are myriad techniques for demographic parity which can be easily extended for conditional demographic parity.
- **[would strengthen the work]** "The eco criterion is individual-level; it requires that each decision is fair... Similar to the ECO criterion, the CF is also individual-level." This makes it sound like these criteria operate at an unusually very fine-grained level, but they just modify distributions of decisions conditional on the inputs to the decision algorithm. While it's common in the fairness literature to characterize these types of criteria as "individual-level" fairness criteria rather than "group" fairness criteria, it's a very strong assertion to say that "each decision is fair." I find this semantically at odds with the fact that the resulting classifiers are randomized, so my concern is that this phrasing will be confusing for other readers.

**Strengths And Weaknesses:**

Strengths
---------
This paper is well written and generally easy to follow. The empirical studies use sensible baseline methods, and they appropriately use both simulated data (where the counterfactuals are known) and a range of datasets that appear frequently in the fairness literature.

Weaknesses
----------
I have concerns about both the technical soundness of the proposed methods as well as their interest for TMLR readers.

Regarding technical soundness, my most serious concern is that Theorem 1, which describes an optimality property of the ECO-fair predictor, appears to be incorrect. The first two lines of the proof in the appendix suggest an assumption that the sensitive feature $S$ and the remaining features $A$ are independent, but they are not independent under the causal graph in Figure 2, which is the causal structure assumed throughout the paper. (It would be a very different setting if they were independent.) If they are not independent, then equation (8) requires $P(S \mid A)$ in place of $P(S)$, and the result in equation (15) is $\int P(\widehat{Y}^{\text{ml}}(S, a))P(S \mid A = a)dS$.

Indeed, it's easy to come up with a toy example that shows that their predictor doesn't minimize the expected KL-divergence as described in Theorem 1; the predictor that does is $\widehat{Y} \sim \text{Bern}(f(a))$, where $f(a) = \int P(\widehat{Y}^{\text{ml}}(S, a))P(S \mid A = a)dS$.

Since the CF-fair predictor takes the ECO-fair predictor of Theorem 1 as a starting point, these issues propagate to the other method and to Theorem 2, which means they are relevant to all the results in the paper.

Regarding interest to TMLR's audience, I'm not sure that the optimality property described in the two theorems is actually ever desirable, and if not then I'm not sure what interest these methods hold.

More specifically, the authors assume that an initial model $f_{\text{ml}}: \mathcal{S} \times \mathcal{A} \mapsto [0, 1]$ is trained to approximate the historical decision process represented in the data. This model induces a randomized classifier $\widehat{Y}^{\text{ml}}(s, a) \sim \text{Bern}(f_{\text{ml}}(s, a))$, and their two theorems say that their ECO-fair and CF-fair predictors minimize the expected KL divergence with respect to $\widehat{Y}^{\text{ml}}(S, A)$. However, it is well known that the optimal classifier is the deterministic Bayes classifier $\widehat{Y}(S, A) = 1[\mathbb{E}[Y \mid S, A] \geq 0.5]$, not a randomized classifier like $\widehat{Y}(S, A) \sim \text{Bern}(\mathbb{E}[Y \mid S, A])$. Since randomized classifiers are never the most accurate possible classifiers, it is unclear what the value is of minimizing divergence from $\widehat{Y}^{\text{ml}}(S, A)$ rather than the underlying decision process. Once again, it is easy to construct toy examples where their ECO-fair classifier is far less accurate than a simple deterministic ECO-fair classifier constructed via Fairness Through Unawareness, so it's at least clear that these methods aren't optimal in terms of accuracy.

---

### Decision · Action_Editors · 2023-02-27

**Recommendation:** Accept with minor revision

**Comment:**

My recommendation is somewhere between "Accept with minor revision" and "Reject" (but encourage resubmission).  Based on the reviews, author responses, discussion, and my own reading of the revised manuscript, I am taking a chance and recommending **Accept with (*significant but let's call it*) minor revision**.  Please see my specific suggestions for improving the manuscript in the **Claims and Evidence** and **Audience** sections above.  Specifically, while in its current form I think Sections 1 - 3 of the paper lack the clarity on ideal world vs real world and FTU vs. the ECO decision-maker, and are thus of limited interest to the TMLR audience, I believe the authors can address this concern in "minor" revision.

**Summary of reviewer discussion/recommendations**

Two of the reviewers recommended reject or leaning reject, while only one recommended "leaning accept".  My recommendation therefore goes against the general inclination of the three reviewers.  Quoting some concerns accompanying the recommendations:

"While I have no remaining concerns about the technical soundness of their results, I remain unconvinced of the paper's interest to TMLR readers. As the authors point out, the "reality of decision making" in the present world is that the protected feature S often affects the unprotected features A, which is part of what gives rise to unfairness. This is a large part of the motivation for this paper, and for many algorithmic fairness methods in general. However, Theorems 1 and 2 provide guarantees about their methods in a world in which S and A are independent, which the authors describe as an "ideal" decision-making process. Furthermore, the authors acknowledge that a fairness-through-unawareness predictor, which satisfies their ECO-fairness criterion, is optimal in the actual world."

"Although the authors addressed all of my direct concerns, I am very concerned by [...] Theorem 1's assumption of independence between A and S, which seems contradictory of the problem setup. Theorem 1's flaw undermines the entire premise of the paper.

The authors responded to this concern by stating that they're explicitly assuming an idealized distribution where A and S are independent. I am unconvinced based on their reasoning and their proof. A satisfactory proof would include the introduction of this idealized world distribution and a separate proof analysis that the reasoning about the idealized world distribution would result in the optimal reasoning for the existing world distribution."

**Audience:**

I agree with concerns raised by the reviewers that optimality in an "idealized" fair world is of questionable value in practice.  The paper would be considerably improved and of greater interest to the TMLR audience if the authors could provide compelling arguments for why ideal world optimality is useful (see "Claims and Evidence" above).  Furthermore, the work would be of greater interest to the audience if there were a more formal discussion of FTU, comparable to that presented for the ECO decision maker.  In the current version of the paper, discussions of FTU appear throughout the manuscript, and the main theorems (1 and 2) feel like they're trying to sweep under the rug the "real world" optimality of FTU by establishing optimality of the authors' proposed method (ECO decision-maker) in an "ideal" world.

**Claims And Evidence:**

Overall the manuscript is clearly written, with suitable examples and appropriate discussion.  I commend the authors on providing a gentle (re-)introduction to counterfactual fairness rather than assuming that interested readers will be fully familiar with this earlier work. The definitions, methods, and technical results are accurate.

**Minor concern**

I do note that it is very confusing to have A be the unprotected features and S be the protected attribute: in much of the fairness literature (including the original counterfactual fairness paper), A is the protected attribute.  I suggest that the authors flip the roles of S and A.  Indeed in the running example, taking $S$ to be the test $S$core and $A$ to be the protected $A$ttribute would create much better alignment.

**Major concerns**

In the initial round of review, reviewers voiced concern with the independence condition (independence of the features A and protected attribute S) under which optimality is established.  As part of this discussion it was pointed out the the fairness through unawarness (FTU) decision-maker would be optimal in the setting where A and S may be dependent (i.e., in the "actual world").  The paper would be considerably clearer and its arguments more easily digestible if the authors further:
1) provided corresponding theoretical results for optimality of FTU in the "actual world";
2) provided convincing arguments for why optimality in the "ideal world" is a desirable criterion; and
3) emphasized earlier (and with appropriate theory, if possible) the property observed in Figure 7 that the FTU model (while also being ECO-fair) can have decisions that are much more correlated with the protected attribute.

In addition, I would suggest splitting out the "ECO-fair" and "CF-fair" parts of Theorems 1 and 2 and the optimality parts to make it clear that the proposed methods are "fair" without the independence condition, but are only provably optimal under the further independence condition.

For (2), the paper would be more convincing if the authors could establish, for instance, conditions under which decisions made by the ECO decision maker bring future versions of the actual world closer to the idealized world, or offer other arguments for why "idealizing" is desirable.

Regarding (3): The most compelling evidence for the ECO decision-maker comes late in the paper, in the final sections of the Empirical studies section.  There the authors show empirically how both FTU and the ECO decision-maker are ECO fair, but the ECO decision-maker produces decisions that are much less correlated with the protected attribute (here, binary gender).  If the authors are able to establish this difference between FTU and ECO more generally, the paper would be much more convincing.  In the absence of such formalism, discussing this difference between FTU and ECO even in the context of a toy model early in the paper would help readers who must currently wait until Section 4 to get (the clearest) answers on ECO decision-maker vs FTU.